# Screening of Sugarcane Proteins Associated with Defense against *Leifsonia xyli* subsp. *xyli*, Agent of Ratoon Stunting Disease

**DOI:** 10.3390/plants13030448

**Published:** 2024-02-03

**Authors:** Xiao-Qiu Zhang, Yong-Jian Liang, Bao-Qing Zhang, Mei-Xin Yan, Ze-Ping Wang, Dong-Mei Huang, Yu-Xin Huang, Jing-Chao Lei, Xiu-Peng Song, Dong-Liang Huang

**Affiliations:** 1Key Laboratory of Sugarcane Biotechnology and Genetic Improvement (Guangxi), Ministry of Agriculture and Rural Affairs/Guangxi Key Laboratory of Sugarcane Genetic Improvement/Sugarcane Research Institute, Guangxi Academy of Agricultural Sciences, Nanning 530007, China; zhangxiaoqiuxhd@163.com (X.-Q.Z.); zbqsxau@126.com (B.-Q.Z.); yanmeixin@gxaas.net (M.-X.Y.); yaheng830619@163.com (Z.-P.W.); huang18260951004@163.com (D.-M.H.); huangyuxin13@163.com (Y.-X.H.); jchlei1130@163.com (J.-C.L.); 2Guangxi South Subtropical Agricultural Science Research Institute, Chongzuo 532415, China; yongjianliang_605@163.com

**Keywords:** sugarcane, endopolygalacturonase, molecular breeding

## Abstract

Sugarcane is the most important sugar crop and one of the leading energy-producing crops in the world. Ratoon stunting disease (RSD), caused by the bacterium *Leifsonia xyli* subsp. *xyli*, poses a huge threat to ratoon crops, causing a significant yield loss in sugarcane. Breeding resistant varieties is considered the most effective and fundamental approach to control RSD in sugarcane. The exploration of resistance genes forms the foundation for breeding resistant varieties through molecular technology. The *pglA* gene is a pathogenicity gene in *L. xyli* subsp. *xyli*, encoding an endopolygalacturonase. In this study, the pglA gene from *L. xyli* subsp. *xyli* and related microorganisms was analyzed. Then, a non-toxic, non-autoactivating pglA bait was successfully expressed in yeast cells. Simultaneously the yeast two-hybrid library was generated using RNA from the *L. xyli* subsp. *xyli*-infected sugarcane. Screening the library with the pglA bait uncovered proteins that interacted with pglA, primarily associated with ABA pathways and the plant immune system, suggesting that sugarcane employs these pathways to respond to *L. xyli* subsp. *xyli*, triggering pathogenicity or resistance. The expression of genes encoding these proteins was also investigated in *L. xyli* subsp. *xyli*-infected sugarcane, suggesting multiple layers of regulatory mechanisms in the interaction between sugarcane and *L. xyli* subsp. *xyli*. This work promotes the understanding of plant–pathogen interaction and provides target proteins/genes for molecular breeding to improve sugarcane resistance to *L. xyli* subsp. *xyli*.

## 1. Introduction

Sugarcane is the most important sugar crop and one of the leading energy-producing crops in the world. Cane sugar accounts for about 87% of the total sugar production in China [1]. *Leifsonia xyli* subsp. *xyli*, the agent of RSD, is a Gram-positive bacterium. *L. xyli* subsp. *xyli* colonizes the vascular bundles of the sugarcane stem, mesophyll cells, apical tissue, and the area around the leaf sheath [2,3]. *L. xyli* subsp. *xyli* adheres to the inside and outside of cell walls, making the cell wall of cane stem and leaves dissolve or break [4,5]. Due to its highly contagious nature and the absence of symptoms, ratoon stunting disease (RSD) caused by *L. xyli* subsp. *xyli* is a significant concern for sugarcane cultivation [6]. RSD poses a significant threat to ratoon crops, causing reduced yields, stalk abnormalities, and delayed harvest [7]. In China, RSD is also a serious disease widely spread in sugarcane planting regions, with an incidence between 48.9 and 100% [8]. To manage RSD, a combination of practices, such as using disease-free seed cane and adopting resistant varieties, are crucial. Hot water treatment can control RSD to some extent, but it comes with high costs and cannot eliminate the pathogen completely [7,9]. 

Breeding of varieties with resistance to RSD is considered the most effective and fundamental approach to controlling RSD on sugarcane. However, due to the fact that sugarcane is a genetically complex allopolyploid plant, it is difficult to combine various favorable traits together through conventional hybrid breeding. Therefore, developing a sugarcane variety resistant to RSD through a conventional breeding program is a challenging task. Molecular breeding is an important method for developing RSD-resistant sugarcane varieties, as it has been successful in improving various traits in multiple crops [10,11,12].

The proteins directly interacting with pathogenic proteins are the ones most likely to participate in defense against the pathogen or be involved in pathogenicity. Endo-PGs, encoded by the *pglA* gene, are essential factors for plant pathogens to colonize the host and degrade pectin in the plant cell walls, resulting in the maceration of host tissues [13,14]. The *pglA* gene has been proved to be a pathogenicity gene in *Pseudomonas solanacearum* [15], *Xylella fastidiosa* [16], and *Xanthomonas oryzae* pv. *oryzae* [17]. Even the endo-PG protein encoded by *pglA* can cause disease. Necrotic lesions are formed after inoculating the proteins SsPG3 and SsPG6 from *Sclerotinia sclerotiorum* on the leaves of *Arabidopsis thaliana* [18]. The leaves of cotton appear chlorotic, and the plants exhibit dwarfism after inoculation with the proteins of VDPG1 and FOVPG1 derived from *Verticillium dahliae* and *Fusarium oxysporum* f. sp. *vasinfectum* [19].

The genome of *L. xyli* subsp. *xyli* has been sequenced in our previous work [4] (https://www.ncbi.nlm.nih.gov/nuccore/LFYU00000000 accessed on 26 May 2023) and by Monteiro-Vitorello et al. [20], as well as by Wang et al. [21]. In the genome, a *pglA* (https://www.ncbi.nlm.nih.gov/gene/2939326#reference-sequences accessed on 26 May 2023), annotated to be an endopolygalacturonase (endo-PG; EC 3.2.1.15), was proposed to be a pathogenicity gene of *L. xyli* subsp. *xyli* by Monteiro-Vitorello et al. [20]. Therefore, the proteins directly interacting with the pglA gene of *L. xyli* subsp. *xyli* will be the best candidate factors against this pathogen or involved in its pathogenicity. The yeast two-hybrid (Y2H) method is the most suitable solution to screen these interacting proteins, since Y2H has been extensively utilized to unravel protein–protein interactions [22] and has helped to reveal the molecular mechanisms underlying the pathogenic processes [23,24]. Thus, in this work, the Y2H system was used to identify the proteins directly interacting with the pglA protein. This approach aims to accelerate the understanding of the mechanisms related to pglA-triggered pathogenicity and to provide target proteins/genes for improving sugarcane resistance to *L. xyli* subsp. *xyli* by molecular breeding.

## 2. Results

### 2.1. Characteristics of pglA Gene and Protein

The open reading frame (ORF) of the *pglA* gene encompasses conserved domains, notably the glycosyl hydrolases family 28 (spanning positions 577 to 1320), and the transcription termination factor Rho (encompassing positions 477-1196 and 951-1460). Furthermore, the pglA protein is anticipated to possess a signal peptide of Sec/SPI. This signal peptide is projected to be cleaved between positions 23 and 24 (Figure 1A,B). The pglA protein shows high similarity with those in other two *L. xyli* subsp. *xyli* strains (Figure 1C). Subsequently, this signal peptide was excised to construct the pglA bait vector.

### 2.2. Toxicity, Autoactivation and Protein Expression of pglA Bait

Colonies were able to grow on the SD/-Trp and SD/-Trp/X media, similar to those on the positive control pGBKT7 medium. However, no colony developed on the SD/-Trp/X/A medium (Figure 2A). This observation suggests that there was no toxicity or autoactivation exhibited by the pglA bait toward yeast cells.

No target protein was expressed in the Y2HGold strain lacking recombinant plasmid. However, a target protein with a size of 22 kDa was expressed in the Y2HGold strain containing the pGBKT7-BD plasmid, and a target protein with a size of 57 kDa was expressed in the Y2HGold strain harboring the pGBKT7-53 plasmid. A target protein with a size of 74 kDa was detected in the Y2HGold carrying the pGBKT7-pglA plasmid. The outcomes affirm the successful expression of the pglA bait in the Y2HGold strain (Figure 2B).

### 2.3. Construction of Y2H Library

The total RNA extraction yielded distinct bands corresponding to the ribosomal 28 S and 18 S (Figure 3A), and the total RNA displayed an OD_260_/OD_280_ ratio of 2.11, suggesting the high quality of the total RNA. Subsequently, cDNA was synthesized using SMART technology (Clontech) (Figure 3B), and purification using a Chroma Spin-1000 column eliminated smaller cDNA fragments (Figure 3C). The purified cDNA was then cloned into the pGADT7 vector to construct a Y2H library. Remarkably, the Y2H library consisted of over 1.0 × 10^6^ primary clones in total, with a final library titer surpassing 4.0 × 10^7^ cfu/mL. The inserted cDNA fragments ranged in size from 0.4 to 2.0 kb (Figure 3D). Impressively, the cDNA library showcased an approximate 100% recombination rate.

### 2.4. Screening of pglA-Interacting Proteins

Under a 1/10,000 dilution, a count of 560 clones emerged on SD/-Leu medium, while more than 2000 clones grew on SD/-Trp medium, and 84 clones formed on SD/-Leu/-Trp medium. This accumulation led to a total of 9.66 × 10^6^ zygotes clones (Figure 4A–C). Upon re-inoculating the blue clones from the SD/-Leu/-Trp/X/A medium onto the SD/-Ade/-His/-Leu/-Trp/X-a-Gal/AbA medium, seven clones surfaced (Figure 4D). PCR amplifications of these clones exhibited a distinct primary band, affirming that the positive clones carried the exclusive AD plasmid (Figure 4D). The colonies of both the positive and negative controls exhibited normal growth, providing confidence in the reliability of our results (Figure 4F,G).

### 2.5. Identification of Proteins Interacting with pglA

The outcomes from both BLAST and UniProt analyses identified a total of six characterized proteins. They included like heterochromatin protein (LHP1), SNF1-related protein kinase regulatory subunit beta-1, histidine-rich calcium-binding protein, DNA-directed RNA polymerase III subunit RPC4, E3 ubiquitin-protein ligase RGLG2, and polyubiquitin-C. Furthermore, one protein was recognized as the yeast vector pDEST-GADT7 (Table 1).

### 2.6. Re-Test of Proteins Interaction

To verify the protein interaction, one protein, 2A (SNF1-related protein kinase regulatory subunit beta-1), was selected for further analysis. The partial sequence of 2A with 619 bp was obtained by sequencing. Then RACE technology was used to obtain the full-length sequence of the 2A gene, SoSnRK1β1 (OP390183). By constructing bait vectors and performing Y2H analysis, the co-cultured strains grew normally with a blue appearance on DDO/X, DDO/X/A, QDO/X/A media, indicating that the pGADT7-SoSnRK1β1 plasmid interacted with the pGBKT7-pglA plasmid (Figure 5A). No false positives of the protein were expressed by the SoSnRK1β1-Y187 strains (Figure 5B), confirming the reliability of the proteins identified in this work.

### 2.7. Gene Expression Analysis

The gene expression of these six proteins was analyzed based on the transcriptome database of sugarcane infected by *L. xyli* subsp. *xyli* [4]. After *L. xyli* subsp. *xyli* inoculation for 60 days, the gene expression levels of 1A and 2A in *L. xyli* subsp. *xyli*-infected plants were higher than those in the control plants. The gene expression level of 3A (c70487_g1) showed no difference between *L. xyli* subsp. *xyli*-infected and the control plants. However, the gene expression levels of 5A, 6A, and 7A were lower in *L. xyli* subsp. *xyli*-infected plants than those in the control plants. After *L. xyli* subsp. *xyli* inoculation for 90 days, the gene expression level of 3A (c70487_g1) did not show a significant difference between *L. xyli* subsp. *xyli*-infected and the control plants. However, the gene expression levels of 1A, 2A, 6A, and 7A in *L. xyli* subsp. *xyli*-infected plants were higher than those in the control plants, whereas the gene expression level of 5A in *L. xyli* subsp. *xyli*-infected plants was lower than that in the control (Figure 6). These results indicate that genes were positively, negatively, or not regulated by *L. xyli* subsp. *xyli* infection, but the interaction happens simultaneously at the protein level. These results also suggest that different levels of regulatory mechanisms participate in the interaction between sugarcane and *L. xyli* subsp. *xyli*.

## 3. Discussion

Sugarcane plays a pivotal role as a major source of sugar and renewable energy, contributing to the global economy. RSD profoundly impacts sugarcane by reducing yields and stalk length, hindering sugar production [25]. Developing sugarcane varieties resistant to RSD stands as the most economically effective approach to mitigate its impact on the sugarcane industry. Identifying RSD-resistant genes is a crucial endeavor in breeding sugarcane varieties resistant to RSD through molecular technology. Ento-PG has been proved to be a pathogenic enzyme in plants [18,19,20]. In *L. xyli* subsp. *xyli*, the agent of RSD in sugarcane, a *pglA* gene also encodes a pathogenic ento-PG. The identification of proteins that directly interact with ento-PG may accelerate the deciphering of the mechanism underlying RSD in sugarcane, providing target proteins/genes for RSD-resistant variety improvement by molecular breeding.

The yeast two-hybrid (Y2H) method is a widely used technique for identifying protein–protein interactions in various organisms, including plants [26]. A high-quality Y2H library is vital for dissecting protein functions and interactions. Factors such as library titer, recombination rate, and inserted fragments size gauge the cDNA library quality [27]. Ideally, the library titer should surpass 1.7 × 10^5^ cfu/mL. In addition, the bait protein, which must be efficiently expressed in the yeast host without causing toxicity or autoactivation of reporter genes, is the focal point of a Y2H experiment [28]. Additionally, the signal peptide is usually removed from the bait protein to prevent improper localization or secretion in the yeast cells. By analyzing the *pglA* gene, it was observed that it had the signal peptide of Sec/SPI. Consequently, the signal peptide was deleted to construct a bait vector. Further work affirmed that the pglA bait protein could be expressed in yeast cells, and no toxicity or autoactivation was exhibited by the pglA bait toward yeast cells. These findings form the foundation for the further identification of pglA-interacting proteins through Y2H. Furthermore, the present study achieved a final library titer exceeding >4.0 × 10^7^ cfu/mL, with inserted cDNA fragments with lengths ranging from 0.4 to 2.0 kb. The recombination rate approached 100%, indicating that a superior-quality library was constructed for further research. Eventually, the presence of six proteins that directly interacted with pglA was successfully identified (Table 1). For instance, six proteins were screened to interact with TpLAP, the leucine aminopeptidase of *Taenia pisiformis* [29]. Similarly, the pore-forming toxin-like gene *PFT*, responsible for conferring resistance against *Fusarium* head blight disease (FHB) in wheat, was found to interact with 23 proteins [24].

Like heterochromatin protein (LHP1) has been found to mitigate abscisic acid (ABA) sensitivity by directly suppressing the expression of the ABA-responsive genes [30]. It also influences the isochorismate synthase 1 (ICS) gene within the salicylic acid (SA) biosynthesis pathway, up-regulating the inactivating enzyme salicylate/benzoate carboxyl methyltransferase (BSMT1), leading to an alteration in salicylic acid content [30]. This subsequently contributes to a decrease in the ABA level in *L. xyli* subsp. *xyli*-infected stalks [4]. Thus, pglA might interact with LHP1 to suppress ABA biosynthesis, potentially inducing stunting in sugarcane, a hallmark of RSD-infected plants [31]. LHP1 is also recognized for its role in regulating plant height, along with influencing leaf and shoot development [32].

SNF1-related protein kinase 1 (SnRK1), the plant ortholog of the yeast sucrose non-fermenting 1 (SNF1) and the mammalian AMP-activated protein kinase (AMPK), constitutes a heterotrimeric complex featuring an α catalytic subunit, a β regulatory subunit, and a γ regulatory subunit [33,34]. SnRK1-mediated signaling profoundly influences the ABA biosynthesis pathway and plant responses to viral, bacterial, fungal, and oomycete pathogens [35,36]. The SnRK1 is activated in response to ABA via SnRK2-containing complexes [36]. In addition, SnRK1 bolsters plant disease resistance through the phosphorylation and destabilization of the WRKY3 repressor [37]. In this work, sugarcane SnRK1 might enhance resistance to *L. xyli* subsp. *xyli* via interaction with pglA, thereby mediating the ABA pathway and immune response. Consequently, SnRK1 emerges as a potential putative primary target for enhancing *L. xyli* subsp. *xyli* tolerance through molecular breeding [35].

Histidine-rich calcium-binding protein (HRC) was initially identified within the QTL *Fhb1* region of wheat [38]. Plants possessing functional *HRC* genes exhibited susceptibility, while those with mutated *HRC* alleles displayed resistance to *Fusarium* head blight (FHB) in wheat [38]. Similarly, silencing the *HRC* gene enhanced multiple disease resistance in potatoes [39]. Additionally, the HRC protein was characterized in *Leymus chinensis* (*LcHRC*), where it modulates abscisic acid (ABA)-responsive gene expression through interaction with the histone deacetylation protein (*AtPWWP3*) [40]. Consequently, HRC negatively correlates with plant disease. Therefore, sugarcane HRC may participate in disease resistance or pathogen pathogenesis by interacting with the pathogenic protein ento-PG.

Ubiquitination governs a myriad of cellular functions in response to biotic and abiotic cues. E3 ubiquitin (Ub)-protein ligase is vital for post-translational modifications (PTMs) that intricately regulate various steps of plant immune signaling [41]. K48 polyubiquitination, a well-studied form, leads to targeted protein degradation through the ubiquitin-proteasome system (UPS) [42]. In *Arabidopsis*, Marino et al. [43] identified the E3 Ub-ligase MIEL1 interacting with the TF MYB30, enhancing resistance responses via down-regulating *MIEL1* to accumulate MYB30 after inoculation with bacteria. Yu et al. [44] demonstrated E3 Ub-ligase EIRP1 interacting with the nuclear TF VpWRKY11, promoting defense in Chinese wild grapevine (*Vitis pseudoreticulata*). *Arabidopsis*’ E3 Ub-ligase RGLG1 and RGLG2 respond to drought stress, interacting with ERF53, with RGLG2 negatively regulating drought responses [45]. The discovery of RGLG2 interacting with pglA in *L. xyli* subsp. *xyli*-infected sugarcane suggests its involvement in bacterial stress response. Additionally, E3 Ub-ligases are known to regulate ABA signaling during abiotic stress [41].

The RNA polymerase III subunit RPC4, a member of DNA-dependent RNA polymerases (Pols) III, plays a role in the transcription of 5S rRNAs and tRNAs. This process is crucial for precise gene transcription and protein synthesis [46,47]. Pol III, the largest RNA polymerase, is highly conserved across eukaryotic organisms [48]. Nguyen et al. [49] discovered that weak or absent *RPC4* expression caused the loss-of-function alleles in rice *DGS1*-nivara^s^ and *DGS2-T65*^s^, leading to hybrid incompatibility. In *N. benthamiana*, Nemchinov et al. [50] demonstrated that *RPC5L* silencing disrupted core Pol III transcripts and diverse cellular processes, including stress responses. Despite classic Pol III genes being considered house-keeping genes, their regulation remains understudied [51]. Likewise, Pol III regulation in sugarcane under *L. xyli* subsp. *xyli* infection, despite RPC4′s interaction with pglA, remains poorly investigated.

## 4. Materials and Methods

### 4.1. Sequence Characteristics of the pglA Gene

The sequence of *pglA* gene (https://www.ncbi.nlm.nih.gov/gene/2939326#reference-sequences accessed on 26 May 2023) was obtained from strain *Lxx*GXBZ01 (LFYU00000000), which was sequenced in our previous work [4] (https://www.ncbi.nlm.nih.gov/nuccore/LFYU00000000 accessed on 26 May 2023). The conserved domains of pglA were analyzed using the Conserved Domain Search Service (CD Search) on NCBI. The SignalP-5.0 (https://services.healthtech.dtu.dk/service.php?SignalP-5.0 accessed on 10 May 2023) was employed to predict the presence of signal peptide. The BLASTp tool (https://blast.ncbi.nlm.nih.gov/Blast.cgi?PROGRAM=blastp&PAGE_TYPE=BlastSearch&LINK_LOC=blasthome accessed on 10 May 2023) on NCBI was used to identify the homologous amino acid sequences from other pathogens.

### 4.2. Construction, Toxicity Testing, Autoactivation and Expression of pglA Bait

The sequence of *pglA*, excluding signal peptide sequence (1–81 bp), was obtained through direct gene synthesis. The resulting fragment was subsequently inserted into the pGBKT7 vector. The recombinant vector, pGBKT7-pglA, was then introduced into *Saccharomyces cerevisiae* strain Y2HGold using Yeastmaker^TM^ Yeast Transformation System 2 (Clontech, CA, USA). To evaluate the potential toxicity and autoactivation of the pglA bait, the transformed cells were plated onto three distinct media of SD/-Trp, SD/-Trp/X, and SD/-Trp/X/A, respectively. Following plating, the cells were cultured at 30 °C for 3–5 days. The strain of Y2HGold containing the pGBKT7-BD vector was spread on an SD-Trp medium, as a positive control.

To determine the expression of pglA bait, the transformed cells were cultured in an SD/-Trp broth medium at 30 °C under 200 rpm for 4–8 h, until the OD_600_ value reached the range of 0.4–0.6. As positive control, the Y2HGold strain carrying the pGBKT7-53 vector and the Y2HGold strain carrying the pGBKT7-BD vector were also cultured in an SD/-Trp broth medium. For negative control, the wild type Y2HGold strain was cultured in a YPDA broth medium. The various yeast cells were harvested via centrifugalization, and the total protein was extracted using the Yeast Protein Extraction Reagent (Takara, Dalian, China).

### 4.3. Construction of Y2H Library

Due to the inability of *L. xyli* subsp. *xyli* to colonize resistant sugarcane varieties, the highly susceptible *Saccharum officinarum* L. cultivar Badila was selected to ensure successful infection and protein interaction. The plants were cultivated in a germplasm repository located at 108°22′, 22°48′ within the Sugarcane Research Institute at Guangxi Academy of Agricultural Sciences, in Nanning, Guangxi, China.

For library construction, the total RNA was extracted from the first internode of stalk above the ground, excluding the epidermis, using the MiniBEST plant RNA extraction kit (Takara, Dalian, China). Subsequently, cDNA synthesis was carried out using the SMART cDNA library construction kit and the Advantage 2 PCR kit (Clontech, CA, USA). The resulting cDNA was then normalized using the TRIMMER-DIRECT cDNA normalization kit (Evrogen, Moscow, Russia) according to the manufacturer’s instructions. The normalized cDNA was further amplified using the cDNA normalization kit and Advantage 2 PCR kit (Takara, Dalian, China). Following amplification, the normalized cDNA was purified using the MiniBEST DNA fragment purification kit (Takara, Dalian, China). To eliminate low-molecular-weight cDNA fragments and small DNA contaminants, the cDNA was excised from a 1% agarose gel after *SfiI* digestion and purified using CHROMA SPIN-1000 columns (Clontech, CA, USA). The purified cDNA was directionally cloned into the pGADT7-SfI vector (a library of prey proteins with the Gal4 activation domain; Clontech, CA, USA) at the *SfiI* A (5′-GGCCATTACGGCC-3′) and *SfiI* B (3′-CCGGCGGAGCCGG-5′) sites to establish the primary cDNA library.

The primary library was transformed into HST08 competent cells through electro-transformation under the condition of 1.8 KV, 200 Ω, 25 μF. The transformed mixture was spread onto LB media supplemented with ampicillin (Amp^+^), followed by overnight incubation at 37 °C until colonies became visible. Subsequently, the transformation efficiency and number of primary colonies were calculated. From the pool of primary colonies, 16 were randomly selected and subjected to PCR amplification using the primer pGADT7-F/R (GGAGTACCCATACGACGTACC/ TATCTACGATTCATCTGCAGC) to assess the insert sizes and the recombination rate of the Y2H library. To validate the normalization results, 96 colonies from the primary library were selected for sequencing. The primary library was the re-transformed into HST08 competent cells, spread across 10 plates with LB medium, and cultured at 37 °C overnight. The plasmids were harvested using the NucleoBond Xtra Midi EF kit (MN, NRW, Germany), and transformed into *S*. *cerevisiae* strain Y187 using the YeastmakerTM Yeast Transformation System 2 (Clontech, CA, USA). The transformed Y187 cells were then plated onto 100 SD/-Leu medium plates and cultured at 30 °C for 3 days. Colonies of transformed Y187 were collected using a freezing medium containing 25% glycerin and stored at −80 °C for further use within the Matchmaker TM gold Y2H system.

### 4.4. Screening of pglA-Interacting Proteins

The pglA-Y2HGold strains were cultured on a solid SD/-Trp medium for 3 days, yielding yeast colonies with a diameter of 2–3 mm. These colonies were co-cultured with the Y2H library strains using liquid SD/-Trp medium in a yeast-mating way, under shaking at 40 rpm and incubation at 30 °C for 24 h. To conduct gradient dilution, a small amount of the co-culture yeast cells was extracted for dilutions of 1/10, 1/100, 1/1000, and 1/10,000, respectively. For each gradient dilution, a 100 μL droplet was spread onto SD/-Leu, SD/-Trp, and SD/-Leu/-Trp media. The remaining dilution was plated on SD/-Leu/-Trp/ X-a-Gal/AbA medium across 50 plates. As a positive control, the pGBKT7-53 plasmid and the pGADT7-T plasmid were co-transformed into the Y2HGold strain. Correspondingly, the pGBKT7-Lam plasmid and the pGADT7-T plasmid were co-transformed into the Y2HGold strain to serve as the negative control. The strains of positive and negative controls were applied to SD/-Leu/-Trp and SD/-Leu/-Trp/ X-a-Gal/AbA media, respectively. The blue colonies were inoculated on an SD/-Ade/-His/-Leu/-Trp/X-a-Gal/AbA medium, and the Matchmaker Insert Check PCR Mix 2 kit was used to amplify the developed clones. The PCR product was purified, followed by sequencing. The sequences were then analyzed using the BLAST (https://blast.ncbi.nlm.nih.gov accessed on 20 July 2023) and Uniprot (https://www.uniprot.org/uniprot/ accessed on 20 July 2023) platforms to unveil protein names and functions. The homology function in *Sorghum bicolor* was determined using BLAST.

## 5. Conclusions

The proteins interacting with pglA are mainly involved in ABA pathways and the immune system, suggesting that sugarcane interacts with *L. xyli* subsp. *xyli* through these two pathways to either cause pathogenicity or develop resistance (Figure 7). This is the first report of protein interactions between sugarcane and pglA from *L. xyli* subsp. *xyli*. This study enhances the understanding of pglA’s role in pathogenicity and provides potential target proteins/genes for molecular breeding to enhance sugarcane’s resistance against *L. xyli* subsp. *xyli*. Future research could focus on validating the roles of these proteins in conferring resistance and elucidating the underlying mechanisms.

## Figures and Tables

**Figure 1 plants-13-00448-f001:**
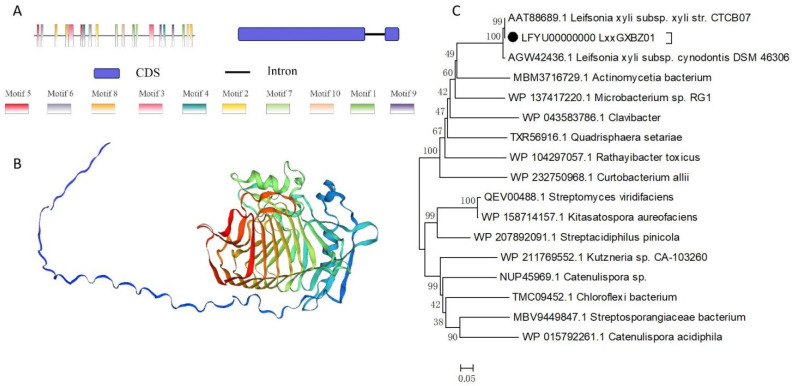
Characteristics of *pglA* gene and protein. (**A**) Schematic structure of gene and protein generated by TBtools-Ⅱ software. (**B**) Tertiary structure of pglA. The colors illustrate the sequence transitioning from the N-terminus to the C-terminus, progressing from blue to red. (**C**) Phylogenetic analysis of pglA in bacterium.

**Figure 2 plants-13-00448-f002:**
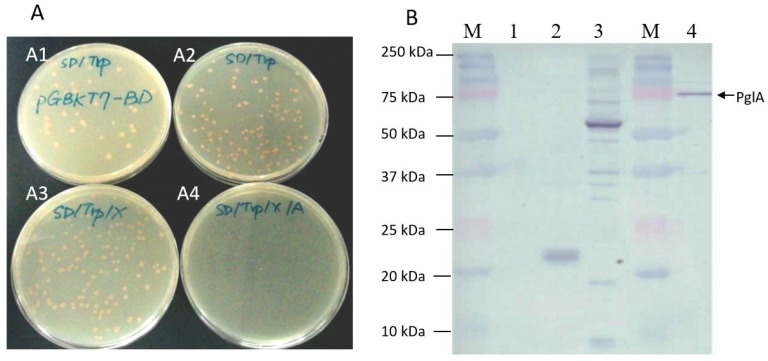
Testing for toxicity, autoactivation and protein expression of pglA bait. (**A**) The protein of pglA bait was expressed in Y2HGold. The strain of Y2HGold, transformed with the pGBKT7-BD vector, was spread on SD-Trp medium (**A1**). The transformed cells were spread on SD/-Trp (**A2**), SD/-Trp/X (**A3**), and SD/-Trp/X/A media (**A4**), respectively. (**B**) Protein expression of pglA bait. M = marker; 1 = Y2HGold without recombinant plasmid; 2 = Y2HGold was transformed with pGBKT7-BD plasmid; 3 = Y2HGold was transformed with pGBKT7-53 plasmid; 4 = Y2HGold was transformed with pGBKT7-pglA plasmid.

**Figure 3 plants-13-00448-f003:**
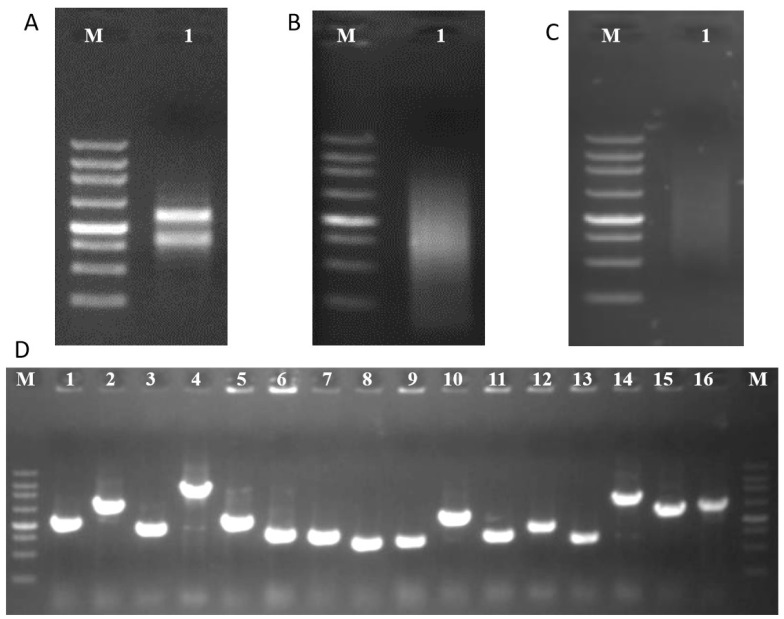
Construction of Y2H library. (**A**) Agarose gel electrophoresis of total RNA from *S. officinarum*. (**B**) cDNA after normalization was evaluated by 1% agarose gel electrophoresis. (**C**) Purified cDNA after eliminating small cDNA fragments using a Chroma Spin-1000 column. (**D**) PCR amplification for the inserted fragments of cDNA library. M = 250 bp DNA ladder (Takara, Dalian, China).

**Figure 4 plants-13-00448-f004:**
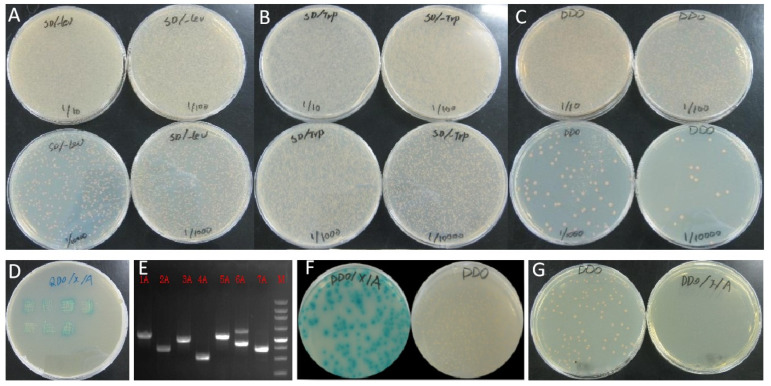
Screening of pglA-interacting proteins. Clones grew on the medium of SD/-Leu (**A**), SD/-Trp (**B**) and SD/-Leu/-Trp (**C**), respectively. Seven clones grew on an SD/-Ade/-His/-Leu/-Trp/X-a-Gal/AbA medium (**D**) and the PCR amplifications of seven clones showed that the main band was distinct (**E**). The colonies of positive and negative controls grew normally (**F**,**G**).

**Figure 5 plants-13-00448-f005:**
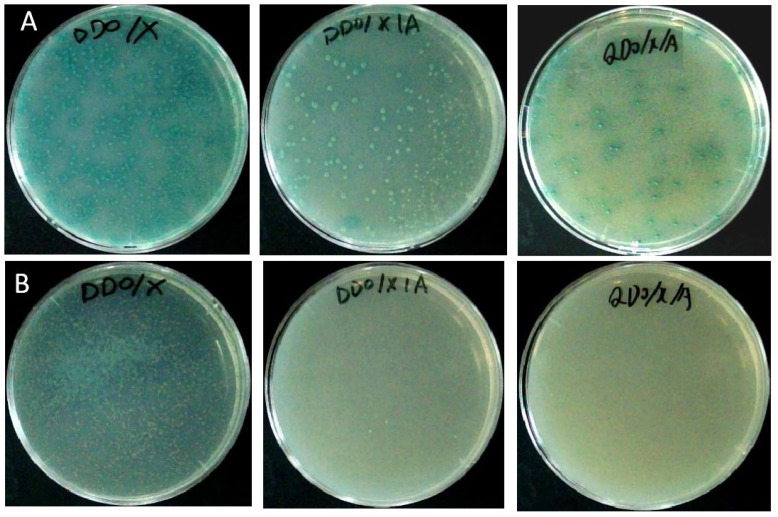
Verification of the interaction between SoSnRK1β1 and pglA. (**A**) The co-cultured strains grew normally with a blue appearance on DDO/X, DDO/X/A and QDO/X/A media. (**B**) The co-culture strains grew normally on DDO/X medium, and two blue colonies grew on DDO/X/A medium. No colonies grew on the QDO/X/A medium.

**Figure 6 plants-13-00448-f006:**
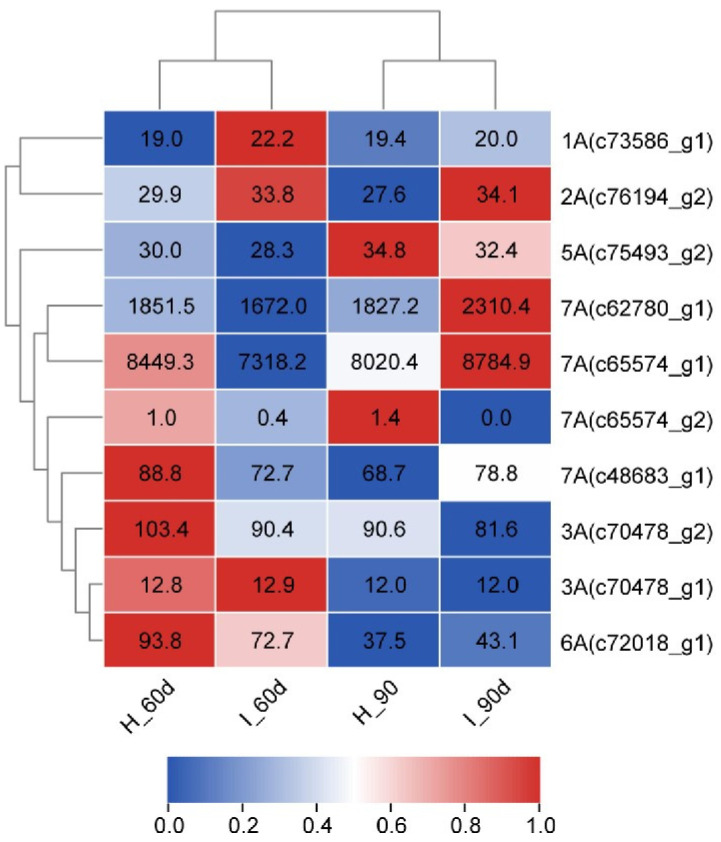
Expression analysis of the proteins encoded by genes identified in this work when infected by *L. xyli* subsp. *xyli*. Heat map drawing based on RPKM values using TBtools software. Different shades of color indicate different levels of gene expression. RPKM values are showed in the box. H_90d: plants inoculated with water after 90 days (control); I_90d: plants inoculated with *L. xyli* subsp. *xyli* after 90 days; H_60d: plants inoculated with water after 60 days (control); I_60d: plants inoculated with *L. xyli* subsp. *xyli* after 60 days.

**Figure 7 plants-13-00448-f007:**
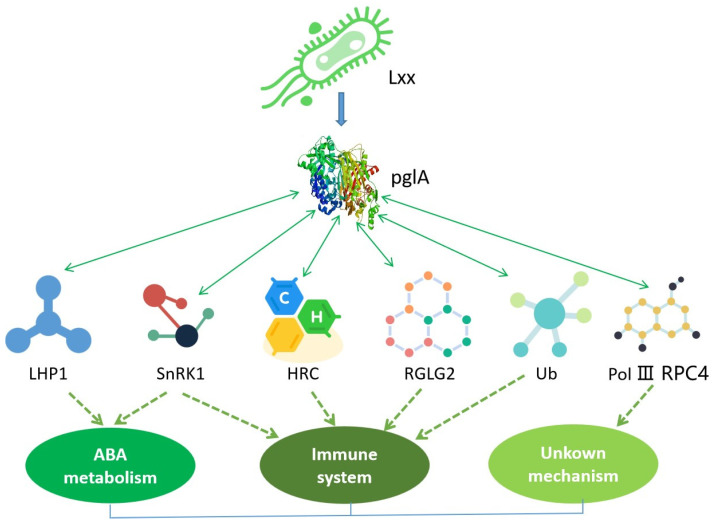
Putative mechanism of sugarcane response to *L. xyli* subsp. *xyli* pglA. Endo-PG, which is encoded by the pglA gene in *L. xyli* subsp. *xyli*, exhibits a direct interaction with sugarcane LHP1, leading to the suppression of the ABA pathway and consequent sugarcane stunting. Additionally, endo-PG also engages in interactions with sugarcane proteins, including SnRK1, HRC, RGLG2, and Ub (ubiquitin), which collectively confer resistance against *L. xyli* subsp. *xyli*. Notably, SnRK1 may also intersect with ABA biosynthesis in response to *L. xyli* subsp. *xyli* invasion. The dynamic interplay between endo-PG and these sugarcane proteins ultimately determines the pathogenicity or resistance of sugarcane.

**Table 1 plants-13-00448-t001:** The information of the proteins interacting with pglA.

No.	Gene Name	UniProtKB Name	Protein Name	Homology Function in *Sorghum bicolor*
1A	AtLHP1	A0A1P8B9G6_ARATH	Like heterochromatin protein (LHP1)	Probable chromo domain protein LHP1
2A	KINB1	KINB1_ARATH	SNF1-related protein kinase regulatory subunit beta-1	SNF1-related protein kinase regulatory subunit beta-1
3A	Hrc	A0A8I5ZZ96	Histidine-rich calcium-binding protein	Sarcoplasmic reticulum histidine-rich calcium-binding protein
4A	/	/	/	Yeast vector pDEST-GADT7
5A	RPC4	RPC4_YEAST	DNA-directed RNA polymerase III subunit RPC4	DNA-directed RNA polymerase III subunit RPC4
6A	RGLG2	RGLG2_ARATH	E3 ubiquitin-protein ligase RGLG2	E3 ubiquitin-protein ligase RGLG2
7A	UBC	UBC_HUMAN	Polyubiquitin-C	Polyubiquitin

## Data Availability

The data that support the findings of this study are available from the corresponding author upon reasonable request.

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
