# Peer review of "Screening of Sugarcane Proteins Associated with Defense against Leifsonia xyli subsp. xyli, Agent of Ratoon Stunting Disease"

_plants, 2024, doi:10.3390/plants13030448_

Round 1

Reviewer 1 Report

Comments and Suggestions for Authors

The manuscript is well written and reads nicely. No major flaws were found but only a few minor comments.

1) In figure 1-B, is the structure predicted or solved experimentally? 

2) In figure 5-A, it looks like the image of QDO/X/A is a bit out of focus. It is hard to tell if there is colony or not. 

3) it would make the paper much stronger if one of the identified protein is characterized in the study. I understand that is probably ongoing work

Author Response

04-02-2024

Dear reviewer,

Many thanks for your positive feedback. Your suggestions and comments greatly helped to improve our manuscript. We have addressed the suggestions in the revised manuscript, and I hope that you will find the revision satisfactory.

The manuscript is well written and reads nicely. No major flaws were found but only a few minor comments.

1) In figure 1-B, is the structure predicted or solved experimentally?

The structure in figure 1B is predicted.

2) In figure 5-A, it looks like the image of QDO/X/A is a bit out of focus. It is hard to tell if there is colony or not.

Thanks for your suggestion. We’ve replaced it with a clearer one.

3) it would make the paper much stronger if one of the identified protein is characterized in the study. I understand that is probably ongoing work

Thanks for your suggestion. The functional characterization of the identified protein will be conducted in the subsequent work.

We hope that we have addressed all the comments/suggestions satisfactorily and the revised version is acceptable for publication.

We look forward to hearing a positive reply from you.

Kind regards,

Dong-Liang Huang, PhD

Professor, Sugarcane Research Institute

Guangxi Academy of Agricultural Sciences, Nanning, Guangxi, China

Reviewer 2 Report

Comments and Suggestions for Authors

The current study deals with an economically important RSD of sugarcane and identifies the potential targets of pgIA gene in sugarcane. The manuscript has some major flaws as highlighted below and requires restructuring.

1. Line 50-51:' no specialized screening procedure': what is a specialized screening procedure? Need to be more specific.

Line 53-54: 'breeding resistant varieties through molecular technology': the context is not clear.

2. 56-68: This paragraph explains the importance of pathogenicity determinants and introduces pgIA. It is important to be brief and to the point.

3. Why was pgIA gene selected out of all the different pathogenicity genes? Does it show higher expression in a particular stage? Why will its interactions prove to be vital for identifying disease-resistance genes?

4. The sequence of pgIA must be submitted as an individual nucleotide accession in the NCBI/DDBJ/EMBLO database.

5. In this study the interactors of pgIA were studied in the library from susceptible Saccharum officinarum L. cultivar Badila variety. So the interactors would be susceptible factors instead of resistance genes as mentioned throughout the manuscript.

6.  In figure 2, the respective protein bands must be labelled by their protein name.

7. From the expression analysis of pgIA target genes, most of the genes showed a decrease in expression in infected samples. However, only 7A was high throughout the infection process.

8. Similar to that in the introduction, the importance of pgIA genes out of all the pathogenicity genes is not clear.

9. In the discussion or result section there is no mention of the pgIA targets (1A, 2A...) and the gene functions discussed. It is important to include a reference to the individual genes' name and their annotation in the manuscript.

10. The current study shows the interaction only through yeast 2 hybrid screening which can be a chance event. I would suggest the use of dilution plating and different concentrations of 3AT to show the strength of interactions. Also verifying these interactions through BiFC, Luciferase, Co-IP or any other means would enhance the findings of this study.

Comments on the Quality of English Language

Extensive editing of the English language is required including spelling mistakes. The figures (e.g. Fig 7) also contain spelling errors.

Author Response

04-02-2024

Dear reviewer,

Many thanks for your positive feedback. Your suggestions and comments greatly helped to improve our manuscript. We have addressed the suggestions in the revised manuscript, and I hope that you will find the revision satisfactory.

The current study deals with an economically important RSD of sugarcane and identifies the potential targets of pgIA gene in sugarcane. The manuscript has some major flaws as highlighted below and requires restructuring.

  1. Line 50-51:' no specialized screening procedure': what is a specialized screening procedure? Need to be more specific.

Line 53-54: 'breeding resistant varieties through molecular technology': the context is not clear.

Thanks for your comments. We removed the confusing sentence about the "specialized screening procedure" and clarified the statement regarding "breeding resistant varieties through molecular technology."

  1. 56-68: This paragraph explains the importance of pathogenicity determinants and introduces pgIA. It is important to be brief and to the point.

Thanks for your suggestion. We revised the statement in this paragraph.

  1. Why was pgIA gene selected out of all the different pathogenicity genes? Does it show higher expression in a particular stage? Why will its interactions prove to be vital for identifying disease-resistance genes?

We are currently simultaneously working on various genes including pglA, stress-inducible protein (Lxx_Rs11070/Lxx22480), cold-shock protein (Lxx_RS08800/Lxx17920), PAT I PROTEIN (Lxx24245 2480407-2481348), etc. The results presented in this work primarily focus on pglA. Following your suggestions in Q10, we plan to employ additional methods such as BiFC, Luciferase, and Co-IP to confirm the interactions for other pathogenicity genes/proteins.

  1. The sequence of pgIA must be submitted as an individual nucleotide accession in the NCBI/DDBJ/EMBLO database.

Yes, the sequence of pglA has been submitted to the NCBI database, and we have included this information in the manuscript along with the link (https://www.ncbi.nlm.nih.gov/gene/2939326#reference-sequences).

  1. In this study the interactors of pgIA were studied in the library from susceptible Saccharum officinarum L. cultivar Badila variety. So the interactors would be susceptible factors instead of resistance genes as mentioned throughout the manuscript.

We have opted to use a susceptible cultivar, Saccharum officinarum L. Badila, to ensure a successful interaction between sugarcane and Lxx. Whether the genes are naturally susceptible or resistant, we can employ techniques such as over-expression or gene editing to modify their expression, aiming to create potential genetically modified (GM) sugarcane with RSD resistance.

  1. In figure 2, the respective protein bands must be labelled by their protein name.

Thanks for your suggestion. The PglA band has been labeled in the updated Figure 2.

  1. From the expression analysis of pgIA target genes, most of the genes showed a decrease in expression in infected samples. However, only 7A was high throughout the infection process.

The transcriptome results show that genes are regulated in various ways (positively, negatively, or not regulated) by L xyli subsp. xyli infection on different days after infection. Simultaneously, interactions occur at the protein level. These findings suggest the involvement of different regulatory mechanisms at various levels in the interaction between sugarcane and L xyli subsp. xyli. This statement has been included in the manuscript.

  1. Similar to that in the introduction, the importance of pgIA genes out of all the pathogenicity genes is not clear.

Due to the abundance of potential pathogenicity genes in Lxx, it is not feasible to discuss all of them in this work. Our focus is specifically on reporting the research related to pglA. The significance of pglA has been emphasized in the Introduction section, in lines 56-66.

  1. In the discussion or result section there is no mention of the pgIA targets (1A, 2A...) and the gene functions discussed. It is important to include a reference to the individual genes' name and their annotation in the manuscript.

The 1A and 2A were annotated as LHP1 and SnRK1, respectively. A detailed discussion on them can be found in the Discussion section, in lines 229-250.

  1. The current study shows the interaction only through yeast 2 hybrid screening which can be a chance event. I would suggest the use of dilution plating and different concentrations of 3AT to show the strength of interactions. Also verifying these interactions through BiFC, Luciferase, Co-IP or any other means would enhance the findings of this study.

Thank you for your suggestion. We plan to explore other methods in our future work. However, for this study, we have re-tested the interaction using the same methods to validate and confirm the results.

  1. Extensive editing of the English language is required including spelling mistakes. The figures (e.g. Fig 7) also contain spelling errors.

Thank you for your suggestion. The manuscript underwent editing by a native English speaker before submission. Additionally, a second English speaker reviewed the revised version.

We hope that we have addressed all the comments/suggestions satisfactorily and the revised version is acceptable for publication.

We look forward to hearing a positive reply from you.

Kind regards,

Dong-Liang Huang, PhD

Professor, Sugarcane Research Institute

Guangxi Academy of Agricultural Sciences, Nanning, Guangxi, China

Round 2

Reviewer 2 Report

Comments and Suggestions for Authors

The authors have attempted to address some of my concerns raised however some of the key issues remain. I am still confused by the rationale of how identifying the targets of pgIA in a susceptible cultivar could help in unravelling disease resistance mechanism. The Lxx utilize this gene to colonize its host and cause infection. So this gene promotes susceptibility not resistance.  It would be important to discuss this in the introduction.

The current study relies on the predominance of Y2H assay only which should be represented robustly with proper controls and dilution plating. As commented previously there is a lack of dilution plating. I am afraid the current interaction study showing colonies in the plate does not provide sufficient information. 

So in this study, the pgIA was selected as an endo polygalacturonase gene that degrades pectin in cell wall. In this context, it is unclear how a cell wall degrading enzyme functions by interacting with signalling and hormonal pathway. The discussion should focus more on elaborating in this context.

In Figure 6, it is unclear what the gene 7A refers to. Here it seems there are 4 isoforms of the gene each having a completely different expression profile. the isoform c65574_g2 expression is negligible, which suggests a pseudogene.

Like my previous comments, the importance of studying pgIA gene must be highlighted at the beginning of the discussion. Currently, the message is not clearly visible.

Author Response

21-01-2024

Dear reviewer,

Many thanks for your positive feedback. Your suggestions and comments greatly helped to improve our manuscript. We have addressed the suggestions in the revised manuscript, and I hope that you will find the revision satisfactory.

The authors have attempted to address some of my concerns raised however some of the key issues remain. I am still confused by the rationale of how identifying the targets of pgIA in a susceptible cultivar could help in unravelling disease resistance mechanism. The Lxx utilize this gene to colonize its host and cause infection. So this gene promotes susceptibility not resistance. It would be important to discuss this in the introduction.

Thank you for your careful review and professional comments. According to your suggestions, we have revised the title to "Screening of Sugarcane Proteins Interacted with PglA, a Pathogenic Protein from Leifsonia xyli subsp. xyli, Agent of Ratoon Stunting Disease".

Furthermore, additional information has been incorporated into the Materials and Methods section (lines 319-321): "Due to the inability of L. xyli subsp. xyli to colonize resistant sugarcane varieties, the highly susceptible Saccharum officinarum L. cultivar Badila was selected to ensure successful infection and protein interaction."

Additionally, we have modified our statement in the introduction section (lines 73-75): "Proteins that directly interact with PglA of L. xyli subsp. xyli represent promising candidates against this pathogen or are involved in its pathogenicity."

The current study relies on the predominance of Y2H assay only which should be represented robustly with proper controls and dilution plating. As commented previously there is a lack of dilution plating. I am afraid the current interaction study showing colonies in the plate does not provide sufficient information.

Thank you for your comments. To validate the reliability of the interaction, we chose one of the identified proteins for a re-test. The results are presented in Results 2.6 Re-test of proteins interaction.

So in this study, the pgIA was selected as an endo polygalacturonase gene that degrades pectin in cell wall. In this context, it is unclear how a cell wall degrading enzyme functions by interacting with signalling and hormonal pathway. The discussion should focus more on elaborating in this context.

Based on current knowledge, the endo-polygalacturonase gene, a cell wall-degrading enzyme, has not been demonstrated to interact with signaling and hormonal pathways. However, our results suggest PglA potentially interacts with the ABA pathway, influencing pathogenicity or resistance. This novel finding requires further validation, as groundbreaking discoveries often emerge unexpectedly and demand additional investigation.

In Figure 6, it is unclear what the gene 7A refers to. Here it seems there are 4 isoforms of the gene each having a completely different expression profile. the isoform c65574_g2 expression is negligible, which suggests a pseudogene. 

Sugarcane is a complex polyploid plant, and the S. officinarum is allo-octoploid with X=10. Consequently, a single protein may be encoded by 8 or even more homologous genes, such as 7A, which corresponds to multiple genes. These genes may belong to different families, and different families of the same gene often exhibit distinctly different functions. Therefore, the expression of these genes is highly complex and may be subject to positive or negative regulation by LXX. In our results, we present gene expression data to illustrate that, even though these proteins interact directly with PglA, the expression of their encoding genes remains intricate. Further research is essential to demonstrate their functions.

Like my previous comments, the importance of studying pgIA gene must be highlighted at the beginning of the discussion. Currently, the message is not clearly visible.

The significance of the pgIA gene has been mentioned in the introduction from lines 59-67. To avoid redundancy, we have briefly reiterated its importance at the beginning of the discussion (lines 204-206).

We hope that we have addressed all the comments/suggestions satisfactorily and the revised version is acceptable for publication.

We look forward to hearing a positive reply from you.

Kind regards,

Dong-Liang Huang, PhD

Professor, Sugarcane Research Institute

Guangxi Academy of Agricultural Sciences, Nanning, Guangxi, China

Round 3

Reviewer 2 Report

Comments and Suggestions for Authors

I accept the manuscripts in its current form.

Author Response

Dear reviewer,

Many thanks for your positive feedback and the time you dedicated to reviewing our manuscript.

Best regards,

Dong-Liang Huang, PhD

Professor, Sugarcane Research Institute

Guangxi Academy of Agricultural Sciences, Nanning, Guangxi, China